# Influence of Low Air Pressure on the Partial Denitrification-Anammox (PD/A) Process

Wen Dai [1], Zhenpeng Han [1], Yongze Lu [1], Shuping Li [2,3], Gangyin Yan [2,3] and Guangcan Zhu [1,2,*]

[1] Department of Environmental Science and Engineering, School of Energy and Environment, Southeast University, Nanjing 210096, China

[2] Key Laboratory of Water Pollution Control and Ecological Restoration of Xizang, National Ethnic Affairs Commission, Xizang Minzu University, Xianyang 712082, China

[3] Information Engineer College, Xizang Minzu University, Xianyang 712082, China

* Correspondence: gc-zhu@seu.edu.cn

**Abstract:** Low air pressure is a feature of high-altitude regions. Domestic wastewater from such regions typically has a low carbon-to-nitrogen ratio (C/N ratio). These factors combine to make traditional biological nitrogen removal in high-altitude regions inefficient and more energy-intensive. The partial denitrification-anaerobic ammonium oxidation (PD/A) process was reported to remove ammonia nitrogen from municipal sewage, consuming fewer carbon sources and requiring no aeration supply. In this study, we set up laboratory-scale reactors in simulated high-altitude environmental conditions, and studied the effect of air pressure on the PD/A process. We found that low pressure promotes nitrogen removal efficiency (*NRE*), achieving 93.0 ± 0.3% at 65 kPa, and the contribution rate of anaerobic ammonium oxidation (anammox) to nitrogen removal increased to 77.7%. Lower dissolved oxygen (DO) concentrations caused by lower air pressure were the reason for higher nitrite accumulation efficiency (*NAE*) in a partial denitrification (PD) system, with measured values of 78.4 ± 2.8% at 65 kPa. The anammox process was promoted by low air pressure, mainly because the low air pressure resulted in higher anaerobic ammonia-oxidizing bacteria activity, with specific anammox activity (*SAA*) reaching 26.3 mg·N/(g·VSS·d). Although the relative abundance of partial-denitrifying bacteria declined slightly, at 65 kPa compared with 96 kPa, they were still the dominant genus of the PD/A sludge, and continued to generate nitrite nitrogen steadily, even at low air pressures. The anaerobic ammonia-oxidizing bacterial abundance remained relatively stable, but their activity was increased, which aided the PD/A process. This study demonstrates how low pressure promotes the PD/A process, indicating the possibility of sustainable improved nitrogen removal in high-altitude regions.

**Keywords:** high-altitude wastewater biological treatment; nitrogen removal; combined partial denitrification and anaerobic ammonium oxidation; low air pressure





## 1. Introduction

The anaerobic ammonium oxidation (anammox) process can directly convert ammonia nitrogen ($NH_4^+$-N) with nitrite nitrogen ($NO_2^-$-N) into nitrogen gas ($N_2$) [1]. Due to its high removal efficiency and low energy consumption in wastewater treatment, anammox is considered to be a promising nitrogen removal process [2–4]. However, a stable supply of $NO_2^-$-N is a bottleneck for mainstream engineering applications of the anammox process.

A high nitrite accumulation and nitrate-to-nitrite transformation rate can be achieved through the partial denitrification (PD) process [5]. Accordingly, a novel integration of the PD and anammox process, namely partial denitrification-anammox (PD/A), was introduced. PD/A has emerged as a promising solution for sustainable nitrogen removal in wastewater [6]. Recent studies [7,8] have shown that the PD/A process requires less aeration and organic resource demand than the traditional nitrification/denitrification process, and is suitable for treating wastewater with low C/N ratios. Reducing greenhouse gas and

excess sludge emissions is another advantage of PD/A over traditional biological nitrogen removal technologies [6]. It also has advantages in low-temperature conditions. Hence, the PD/A process might be suitable for application in high-altitude regions. Nevertheless, uncertainty surrounds the influence of air pressure, particularly low air pressure, on the PD/A procedure.

Due to the low air pressure in high-altitude regions, wastewater produced there has a low saturation of dissolved oxygen (DO), making it difficult for oxygen from the air to transfer into the water [9] and then promote anaerobic reactions. According to [10], a drop in air pressure increased the activity of nitrite-dependent denitrifying bacteria and nitrite-oxidizing bacteria, while decreasing the activity of ammonia-oxidizing bacteria. A study of the operation of combined partial nitrification and anammox under low air pressure showed that the activity of anaerobic ammonia-oxidizing bacteria was promoted [11]. These previous studies suggest that air pressure can have an effect on microorganisms, which may be related to the concentration of dissolved oxygen caused by the air pressure and the related activity of functional microorganisms. Denitrification is a process of converting $NO_3^--N$ to $N_2$ step by step (Figure 1), while in the PD process, $NO_3^--N$ reduction terminates with $NO_2^--N$, and no further denitrifying steps result in a high nitrite accumulation efficiency (*NAE*) [12]. Thus, a stable PD process, that is, a stable *NAE*, is very important for achieving efficient PD/A at low air pressure; however, there are few reports currently available describing this proposed process. Therefore, it is necessary to determine if low air pressure influences the PD/A process, and to confirm whether employing the PD/A process in high-altitude regions is practicable.

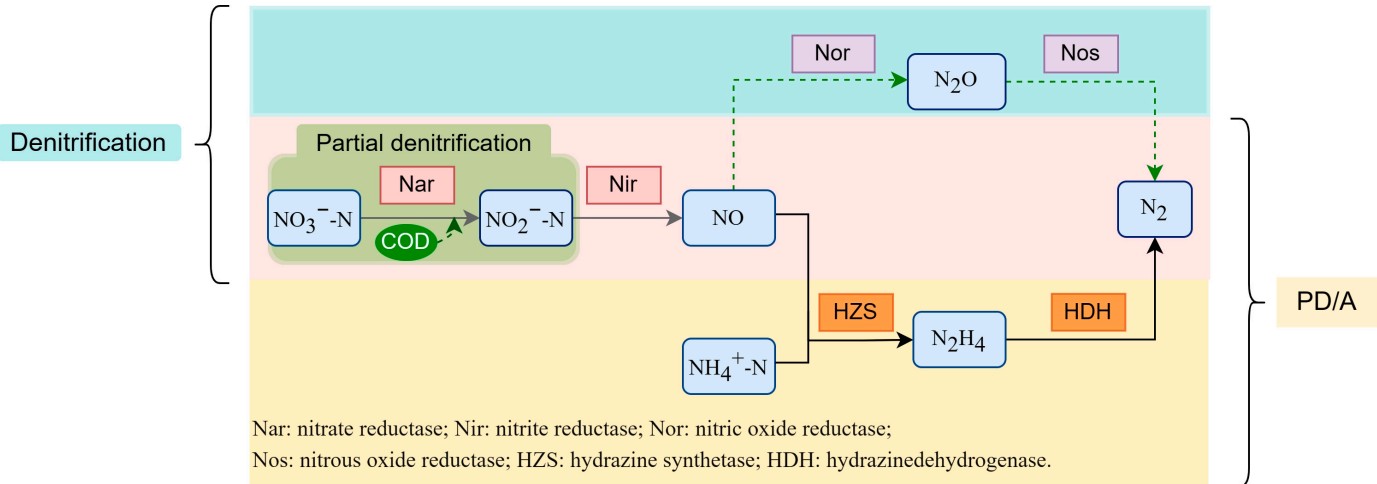

**Figure 1.** Schematic illustration of the denitrification process, including the partial denitrification-anammox (PD/A) process steps.

In this study, two sequencing batch reactors (SBR) and an up-flow solid reactor (USR) were constructed. The three reactors were operated in a simulated high-altitude environment under conditions of varying air pressure. One SBR was used to study the effect of air pressure on the PD process, one on the anammox process, and the USR was used to culture sludge for the PD/A system. In order to research the impacts of air pressure variations on the nitrogen removal performance and pathways in the PD/A process, the nitrogen removal performance of the reactors, ex situ sludge activity, and microbial community structures were evaluated. The study aims to provide a reference for future research and application of the PD/A process and practical experience for sustainable wastewater treatment in high-altitude regions.

## 2. Materials and Methods

### 2.1. Reactor Setup

Partial-denitrifying bacteria were enriched and cultured in an SBR (SBR-PD), while anaerobic ammoxidation bacteria were cultured in a second SBR (SBR-A). The two bacteria were subsequently combined in a USR (USR-PD/A) and used to start a PD/A system.

All the reactors were made of plexiglass with a pH probe and an oxidation–reduction potential (ORP) probe installed in each reactor to monitor the operating conditions over time. Each SBR had a working volume of 3 L (Supplementary Figure S1A) and was controlled by several sets of timing switches to control water inflow and outflow. SBR-A was covered with a black plastic film to block the light.

USR-PD/A had a working volume of 4.5 L with a total height of 100 cm (Supplementary Figure S1B). It consisted of two compartments, a flow-rising area, and a sludge-settling area. For biological attachment and growth, the flow-rising area was filled with biological stuffing. An air diffuser connected to a gas circulating pump was placed at the bottom of the reactor to aerate the sludge with nitrogen to make it appear fluid, to increase the contact area with the biological filler, and to promote sludge film formation on the fillers.

All the reactors were placed in an enclosed modular laboratory. The main body of the modular laboratory was composed of an experimental module, a buffer compartment, an equipment room, and a control room. The modular laboratory had dimensions of 8 m (length) × 2 m (width) × 2.3 m (height) (Figure 2). The experimental module provided simulated environmental conditions for the three reactors. The buffer compartment provided a space for personnel entering the laboratory module to adapt to changes in air pressure.

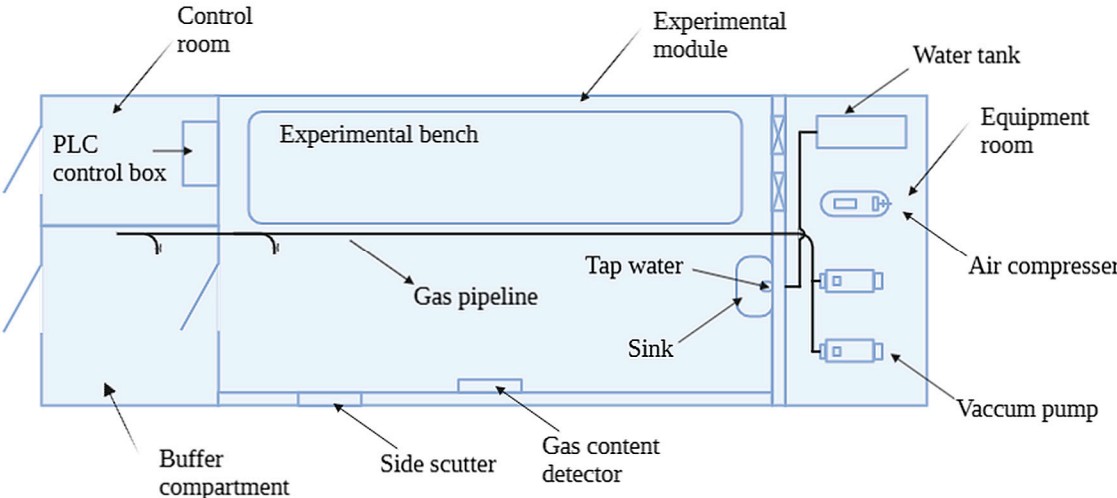

**Figure 2.** Schematic diagram of the high-altitude environment enclosed modular laboratory.

A vacuum pump and the inlet and exhaust valves were controlled using programmable logic controllers (PLCs). When the air in the experimental module was higher than the set pressure, the air in the module was extracted by the PLC-controlled vacuum pump, and the pressure in the experimental module was reduced until the required pressure was achieved. When the air pressure in the experimental module was lower than the required pressure, the PLC-controlled valve opened so that the pressure in the experimental module rose to the set pressure. In addition, the temperature and operational times of the experimental module were also controlled using PLCs.

### 2.2. Seeding Sludge and Synthetic Wastewater

SBR-PD was inoculated with seed sludge taken from a municipal wastewater treatment plant (Xianyang, China). Anammox seed sludge was purchased from the Jiayi Environmental Protection Technology Co., Ltd. (Lianyungang, China) and cultured in SBR-A. Sludge was taken from the two SBRs and combined in the USR-PD/A to initiate the

PD/A process with a mixed liquor suspended solids value of 1500 mg/L. The composition of the synthetic wastewater used in the three reactors is given in Supplementary Table S1. Its pH value was between 6.5 and 7.0.

### 2.3. Operating Conditions

All the reactors were placed into the high-altitude environment experimental module and were initiated at a temperature of $20 \pm 1$ °C and an air pressure of 96 kPa. SBR-PD remained in operation for 180 days, while SBR-A operated for 140 days, and USR-PD/A for 140 days.

During the start-up period, the C/N ratio, the sludge retention time (SRT), and the reaction time were adjusted to obtain a stable SBR-PD state. After 120 days of adjustment, the conversion rate of $NO_3^-$-N to $NO_2^-$-N was stable at about 73%. It took 35 days for SBR-A to reach a stable anammox rate through reaction time adjustment. USR-PD/A was operated for 50 days to achieve a stable denitrification rate of about 90% by adjusting the hydraulic retention time (HRT); by this time, the biofilm had covered the biological fillers.

After the start-up period, three operation phases were conducted by changing the air pressure: Phase 96 kPa, Phase 72 kPa, and Phase 65 kPa. The other operation conditions of the three reactors are summarized in Supplementary Table S2. The effluent samples from all of the reactors were taken daily to investigate the treatment efficiency, and sludge samples from USR-PD/A were collected at the end of the different pressure phases to investigate the microbial community structure.

### 2.4. Batch Tests

Three groups of repeated intermittent experiments were conducted to study the effects of low pressure on PD and anammox. In the PD batch test, a differential pressure meter was connected at the top of each airtight glass vial. Some 40 mL of partial denitrification sludge and 250 mL of synthetic wastewater were added to each vial. The composition of the synthetic wastewater is described in Section 2.2.

After the PD sludge was taken from the end of the SBR-PD drainage stage, the collected sludge was washed with ultra-pure water three times to remove any residual substrate. An anaerobic environment was created in the vials using nitrogen stripping (99.99% pure). The air pressures in the vials were set to 96, 72, and 65 kPa. The $N_2$ in the headspace was then extracted using a syringe, and the gas pressure was measured with a gauge. The vials were put on a magnetic agitator and rotational speed was set to 60 r/min. Three parallel experiments were performed under each set pressure.

The procedure for the anammox batch test was roughly the same as that of the PD experiment. All experiments were performed in the dark.

### 2.5. Analytical Methods

The concentrations of $NH_4^+$-N, $NO_3^-$-N and $NO_2^-$-N were obtained using the APHA method [13], and hemes C was quantified using pyridine hemoglobin spectrometry [14]. These indexes were measured using the Multiskan spectrum (Multiskan Sky, Thermo Fisher, Waltham, MA, USA).

COD concentration was measured using a quick analysis device (Lianhua Tech., Beijing, China). Hydrazine was measured using the $Na_2S_2O_3$ titration method [15]. The temperature, ORP, and pH were analyzed with ORP and pH probes (Sinomeasure Tech., Hangzhou, China).

Extracellular polymeric substances (EPS) were extracted using the heat extraction procedure [16]. The polysaccharide (PS) and protein (PN) of the EPS were measured using the anthrone–sulfuric acid method and a modified Bradford protein assay kit (Shanghai Sangon Biotechnology Co., Ltd., Shanghai, China).

### 2.6. Calculations

The specific anammox activity (*SAA*, mg·N/(g·VSS·d)) was calculated using Equation (1), and the nitrate reduction rate (*NRR*, mg·N/(g·VSS·d)) was calculated using Equation (2):

$$SAA = \frac{NH_4^+{}_{Inf} - NH_4^+{}_{Eff}}{\Delta t \cdot VSS} \tag{1}$$

$$NRR = \frac{NO_3^-{}_{Inf} - NO_3^-{}_{Eff}}{\Delta t \cdot VSS} \tag{2}$$

where *VSS* is the concentration of anammox sludge in a vial (g·VSS/L), and $\Delta t$ indicates the duration of the anammox reaction (days).

The nitrogen removal efficiency (*NRE*, %) was calculated using Equation (3), and the *NAE* (%) was calculated using Equation (4):

$$NRE = \left(1 - \frac{NH_4^+{}_{Eff} + NO_3^-{}_{Eff} + NO_2^-{}_{Eff}}{NH_4^+{}_{Inf} + NO_3^-{}_{Inf}}\right) \times 100\% \tag{3}$$

$$NAE = \frac{NO_2^-{}_{Eff} - NO_2^-{}_{Inf}}{NO_3^-{}_{Inf}} \times 100\% \tag{4}$$

where $NH_4^+{}_{Inf}$, $NO_3^-{}_{Inf}$, $NO_2^-{}_{Inf}$, $NH_4^+{}_{Eff}$, $NO_3^-{}_{Eff}$, and $NO_2^-{}_{Eff}$ are the inflow and effluent concentrations of $NH_4^+$-N, $NO_3^-$-N, and $NO_2^-$-N in the reactors, respectively (mg/L).

### 2.7. Microbial Community Structure Analysis

Activated sludge samples were taken at the end of the 96 kPa phase, 72 kPa phase, and 65 kPa phase of the PD/A system in order to examine the change in the structure of the microbial structure. Genomic DNA was extracted from the sludge samples using cetyltrimethylammonium bromide (CTAB). The sequencing region of the activated sludge DNA was amplified using polymerase chain reaction (PCR), using specific primers with barcodes, Phusion® High-Fidelity PCR Master Mix with GC Buffer (New England Biolabs Inc., Ipswich, MA, USA), and high-fidelity DNA polymerase. The PCR products were sequenced on a PacBio platform. Both PCR and sequencing analyses were conducted by Beijing Novogene Technology Co., Ltd. (Beijing, China).

## 3. Results

### 3.1. PD Process Performance

The denitrification performance of SBR-PD during the start-up period is summarized in Supplementary Figure S2. The results show that the PD process was effectively established, and the *NAE* reached 73.2%, by the end of the start-up period. In general, we discovered that low air pressure was beneficial to the PD process in SBR-PD due to the $NO_2^-$-N concentration, and the *NAE* in the effluent increased with decreasing pressure (Figure 3a).

Effluent from the reactor at the end of Phase 96 kPa, Phase 72 kPa and Phase 65 kPa (i.e., day 10, day 35 and day 60, respectively) was taken for nitrogen concentration analysis. The $NO_2^-$-N concentrations in the three phases were 36.5 ± 0.1 mg/L, 38.6 ± 0.1 mg/L and 39.7 ± 0.2 mg/L, respectively, and the average *NAE* values were 72.5 ± 4.2%, 75.9 ± 5.4% and 78.4 ± 2.8%, respectively. $NO_3^-$-N concentrations at all three pressures were about 0.5 mg/L.

The variations of $NO_2^-$-N and $NO_3^-$-N concentrations in a single cycle of SBR-PD, operated at the three different air pressures, can be seen in Figure 3b,c, respectively. With the decrease in air pressure, the $NO_2^-$-N accumulation rate was faster in the initial 20 min, and the ultimate $NO_2^-$-N concentration was higher, suggesting that PD was not only encouraged, but also the reaction rate was accelerated by lower air pressure. These phenomena above indicate that the conversion rate from $NO_3^-$-N to $N_2$ in the system was reduced, which means that the denitrification was suppressed by the lower air pressure.

The COD concentrations were $32.0 \pm 1.3$ mg/L, $30.4 \pm 1.3$ mg/L and $29.3 \pm 0.8$ mg/L for Phase 96 kPa, Phase 72 kPa and Phase 65 kPa, respectively, which means that the system consumed more COD with a lower air pressure. The $NO_2^-$-N accumulation rate and carbon source consumption during a cycle increased for lower air pressure, which was presumed to be associated with the higher activity of partially denitrifying bacteria.

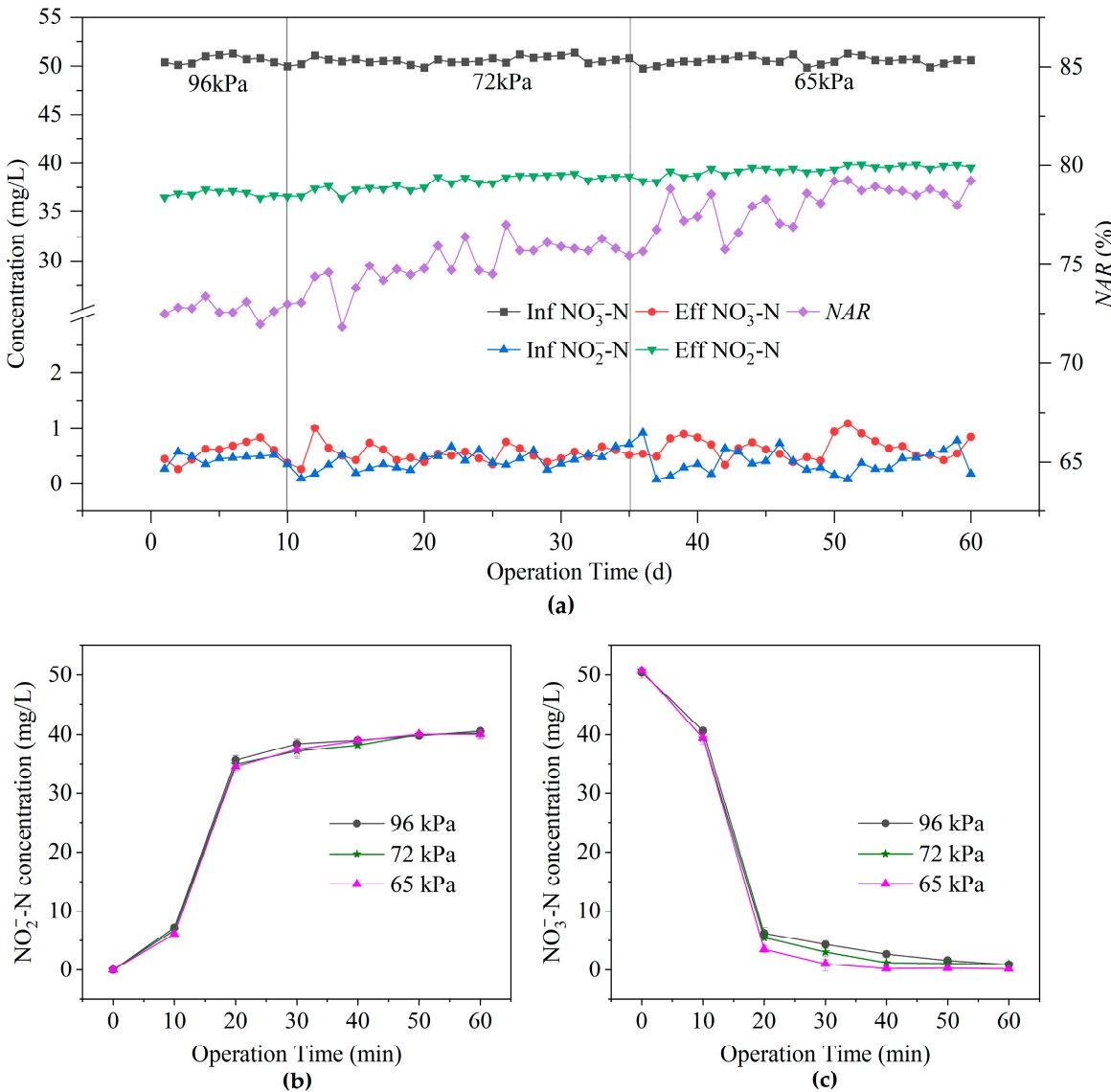

**Figure 3.** Nitrogen conversion performance of SBR-PD against operation time (**a**) for three different air pressure phases. The concentration of (**b**) $NO_2^-$-N and (**c**) $NO_3^-$-N for single cycles of SBR-PD. Inf—inflow. Eff—effluent.

To assemble and withstand adverse environmental conditions, bacteria require EPS, a marker of microbial activity [17]. At the end of each air pressure phase in the reactors (i.e., day 10, day 35, and day 60), samples were taken for EPS analysis in order to better understand how air pressure affects microbial activity during the PD process (Figure 4). The EPS content of the PD sludge was 103.8 mg/(g·VSS) at 96 kPa, and the EPS contents of the sludge at 72 kPa and 65 kPa were about 1.8 times that at 96 kPa. In addition, the PN/PS ratio for both the 96 kPa and 72 kPa air pressure was about 2.0, while for 65 kPa, it was 2.4. It can be assumed that the lower air pressure did increase the EPS content and change its composition. By facilitating full contact between bacteria and inorganic nutrients through adsorption and accumulation, the PS facilitated the conversion of $NO_3^-$-N to

$NO_2^-$-N, which in turn promoted the PD process [11]. Although the PS content at 65 kPa was slightly lower than that at 72 kPa, it was also about 75% higher than at 96 kPa, which is not inconsistent with the *NAE* increasing with lower air pressure.

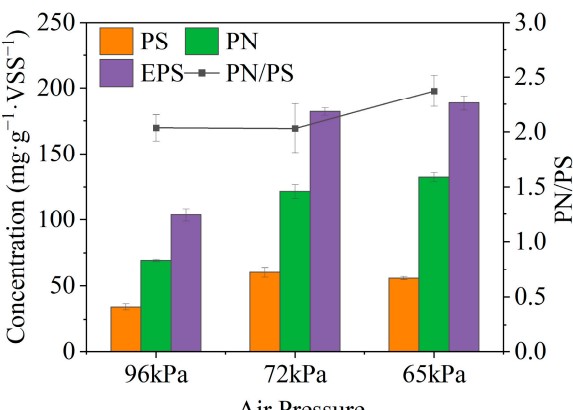

**Figure 4.** Variations in EPS during the PD process at three different air pressures.

Lower air pressures resulted in lower DO in the SBR-PD. The DO concentrations in the system at 96 kPa, 72 kPa and 65 kPa were 0.4 mg/L, 0.3 mg/L and 0.2 mg/L, respectively. In order to investigate whether varied DO concentrations in a reactor at different air pressures are the factor that influences the PD process, 60 min batch tests were conducted at the three pressures, but with the same DO ($\leq$0.05 mg/L); the results are shown in Figure 5.

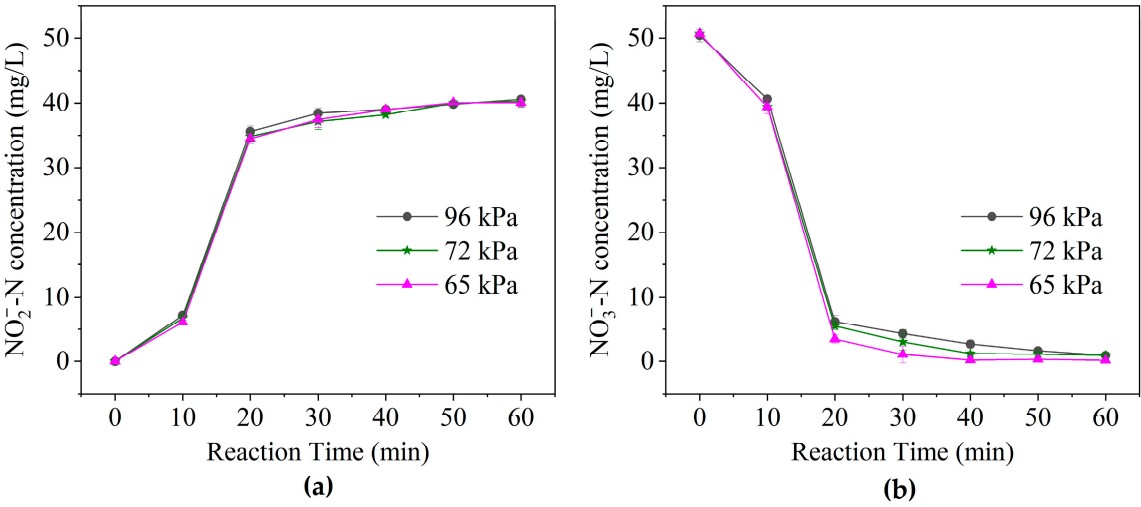

**Figure 5.** The concentration of (**a**) $NO_2^-$-N and (**b**) $NO_3^-$-N of PD batch tests at different air pressures, but with the same low DO concentration ($\leq$0.05 mg/L).

There were no significant differences in $NO_2^-$-N accumulation between the air pressure phases, indicating that the pressure itself had little effect on the PD system. In conclusion, while the pressure itself had little impact on the PD system, it is hypothesized that lower air pressure in the reactor causes lower DO, which promotes PD and increases its rate, leading to an increase in the rate and accumulation of $NO_2^-$-N.

### 3.2. Anammox Process Performance

We found that lower air pressure was beneficial to the anammox reaction in SBR-A. Cycles from 33–35 d (96 kPa), 63–65 d (72 kPa) and 93–95 d (65 kPa) were used for analysis, and *SAA* values were 21.5, 25.1, and 26.3 mg·N/(g·VSS·d), respectively; *NRR* values were 5.7, 6.6, and 6.9 mg·N/(g·VSS·d), respectively.

The ratio of *SAA* and *NRR* at each pressure is equal to the stoichiometric number of $NH_4^+$-N and $NO_3^-$-N in the anammox reaction equation (Equation (5)) [18]:

$$1NH_4^+ + 1.32NO_2^- + 0.066HCO_3^- + 0.13H^+ \longrightarrow$$
$$1.02N_2 + 0.26NO_3^- + 0.066CH_2O_{0.5}N_{0.15} + 2.03H_2O$$

(5)

The *SAA/NRR* ratio was 1:0.26 for the three air pressures. It is therefore clear that anammox was fully functional in the reactor at all three pressures [19].

$NH_4^+$-N, $NO_2^-$-N and $NO_3^-$-N concentrations of SBR-A are presented in Figure 6 at 96 kPa (Figure 6a), 72 kPa (Figure 6b), 65 kPa (Figure 6c). The lower the air pressure, the lower the concentrations of $NH_4^+$-N and $NO_2^-$-N in the effluent, while the $NO_3^-$-N concentrations increased. Compared with those at 96 kPa, the $NH_4^+$-N and $NO_2^-$-N concentrations in the effluent at 65 kPa were lower by 45% and 56%, while the $NO_3^-$-N concentration was greater by 3%.

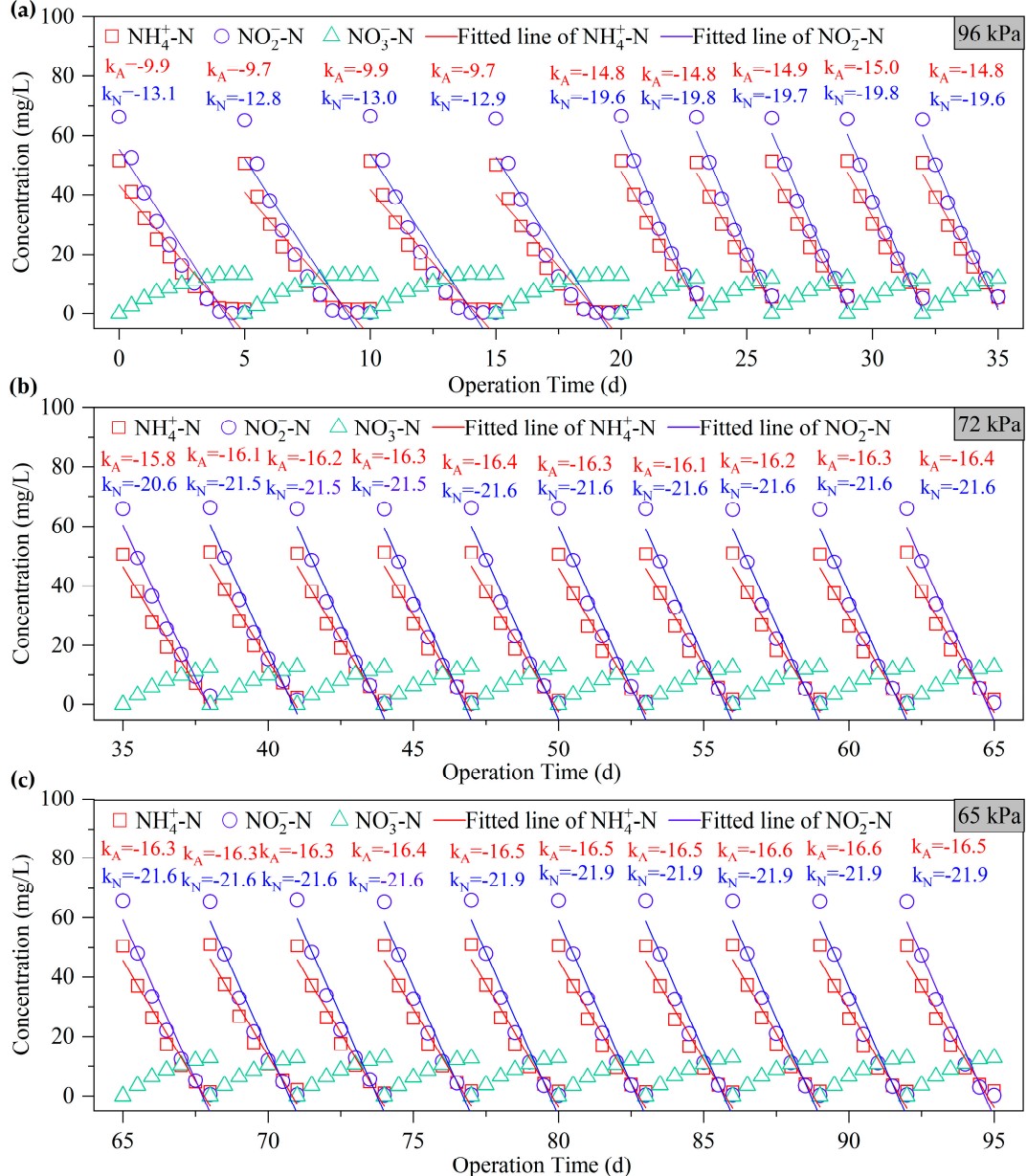

**Figure 6.** $NH_4^+$-N, $NO_2^-$-N and $NO_3^-$-N concentrations of SBR-A at 96 kPa (**a**), 72 kPa (**b**), and 65 kPa (**c**). $k_A$—the slope of the $NH_4^+$-N concentration fitted line. $k_N$—the slope of the $NO_2^-$-N concentration fitted line.

Similar results were obtained from SBR-PD. For low air pressure, the DO concentration in the SBR-A system was low as well, with values of 0.4 mg/L, 0.3 mg/L, and 0.2 mg/L at 96 kPa, 72 kPa, and 65 kPa, respectively. DO may hinder the activity of anammox bacteria [20]; hence, the varied concentration of DO in the reactor at different air pressures is a factor that cannot be disregarded. Therefore, a 12 h batch test was performed to ascertain the impact of air pressure on the anammox process, with the same DO concentration ($\leq 0.05$ mg/L) in the vials. Residual nitrogen concentrations and *SAA* values at 96 kPa, 72 kPa, and 65 kPa are presented in Table 1. The $SAA_{12h}$ at 65 kPa was 52.1 mg·N/(g·VSS·d), which was 1.4 and 1.1 times that at 96 kPa and 72 kPa, respectively, indicating that the anammox efficiency was higher at lower air pressures.

**Table 1.** *SAA* and $NH_4^+$-N concentrations during anammox batch tests with DO concentration held constant ($\leq 0.05$ mg/L).

| Group | $SAA_{12h}$ (mg·N·g$^{-1}$·VSS$^{-1}$·d$^{-1}$) | $NH_4^+$-N Concentration (mg/L) | | | | | | |
|---|---|---|---|---|---|---|---|---|
| | | 0 h | 2 h | 4 h | 6 h | 8 h | 10 h | 12 h |
| 96 kPa | 38.64 | 30.17 | 23.36 | 18.21 | 13.77 | 9.97 | 6.55 | 3.52 |
| 72 kPa | 46.56 | 29.90 | 22.62 | 16.97 | 12.08 | 7.90 | 4.33 | 1.24 |
| 65 kPa | 52.08 | 30.01 | 22.72 | 16.81 | 11.67 | 7.30 | 3.52 | 0.48 |

Hydrazine is the intermediate product of the anammox reaction [21], and its production and transformation significantly affect the activity of the anammox reaction. In the anammox process, $NO_2^-$-N is converted to NO or hydroxylamine by nitrite reductase, then NO and $NH_4^+$-N are converted to hydrazine under the action of hydrazine synthetase; finally, hydrazine is converted to $N_2$ under the action of ammonia oxidase. The results of the batch tests show that at the lower air pressure, the hydrazine concentration in the vials was also lower at every point (Figure 7a). On the one hand, the low air pressure might inhibit the production of hydrazine, but on the other hand, it might encourage hydrazine consumption.

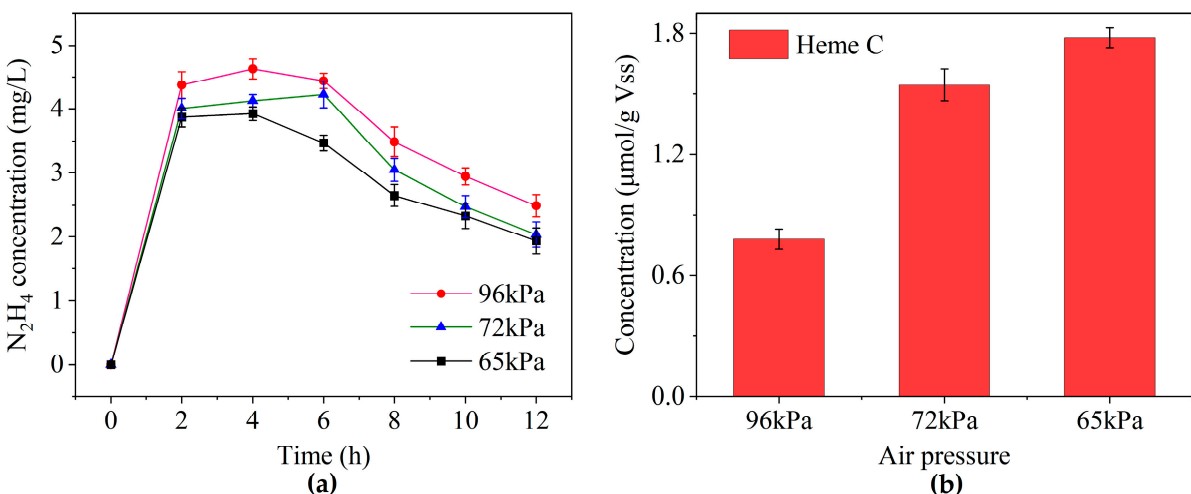

**Figure 7.** Hydrazine (**a**) and heme C (**b**) concentrations during anammox batch tests.

At the same time that hydrazine concentrations were monitored, the heme C contents of the anammox sludge in the vials were also determined in order to quantify the activity of the sludge and to attempt to explain the change in the hydrazine. Previous studies have shown that the chromaticity of anammox sludge is proportional to its activity [22,23], and the chromaticity of sludge is positively correlated with the concentration of intracellular heme C; thus, the activity of anammox sludge can be characterized by heme C concentration.

The heme C concentrations at 96 kPa, 72 kPa and 65 kPa were 0.8 μmol/(g·VSS), 1.5 μmol/(g·VSS), and 1.8 μmol/(g·VSS), respectively. At lower air pressures, the increase in sludge heme C concentration reflected an enhancement in anaerobic ammonia oxidation sludge activity (Figure 7b), indicating that low air pressure promoted the consumption of hydrazine, which was beneficial to the conversion of hydrazine to nitrogen, and finally promoted the anammox reaction. Previous research [11] has found that lower pressure is conducive to improving the activity of anammox bacteria, which is consistent with the results of this study.

### 3.3. Nitrogen Removal Performance in the Combined PD/A System

Nitrogen removal during the PD/A process may be facilitated by low air pressure. Stable operation of the PD/A process in USR-PD/A was achieved by adjusting the hydraulic retention time to 12 h (Supplementary Figure S3). The air pressure of the reactor was adjusted to 96 kPa, 72 kPa, and 65 kPa, and the nitrogen concentration of the effluent is shown in Figure 8. With the decrease in air pressure, the NRE of the system increased gradually, and the average nitrogen removal efficiencies were 89.4 ± 0.4%, 91.4 ± 0.8% and 93.0 ± 0.3% during Phase 96 kPa, 72 kPa and 65 kPa, respectively.

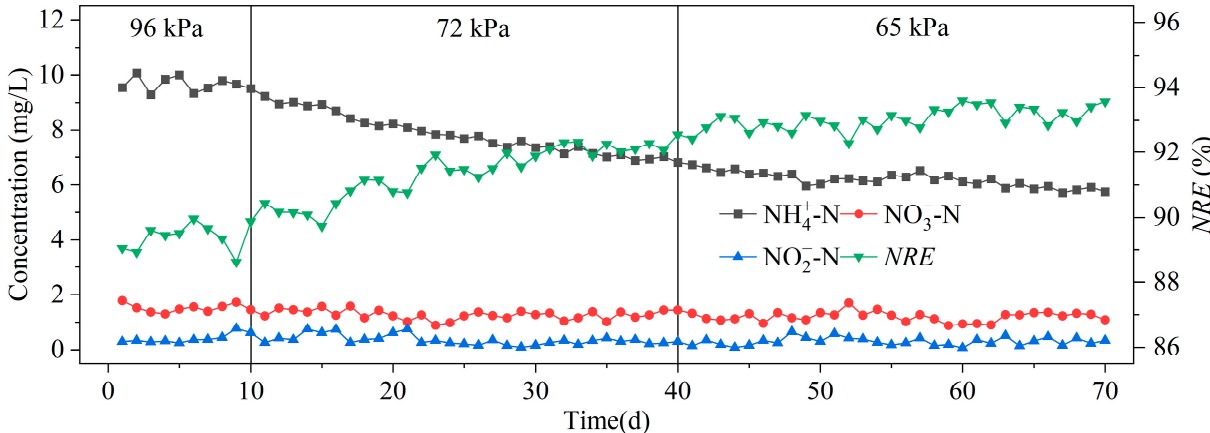

**Figure 8.** Nitrogen removal performance of PD/A process in USR-PD/A at different air pressures.

The nitrogen concentrations in the effluent during the stable operation of the reactor during Phase 96 kPa, 72 kPa and 65 kPa (i.e., day 10, day 40 and day 70) were taken for analysis. The $NH_4^+$-N concentrations in the effluent were 9.7 ± 0.3 mg/L (96 kPa), 7.8 ± 0.7 mg/L (72 kPa) and 6.2 ± 0.3 mg/L (65 kPa). The $NO_2^-$-N and $NO_3^-$-N concentrations varied little between the different air pressures, with a concentration range of about 0.15–0.3 mg/L and 0.9–1.8 mg/L, respectively.

The nitrogen removal contribution proportion and the flow direction of $NO_2^-$-N during a single reaction cycle were analyzed. The reaction equation of anammox is presented in Equation (5). It may be assumed that the $NH_4^+$-N consumption, $NO_2^-$-N consumption and $NO_3^-$-N production during the anammox process were consistent with the theoretical values of 1, 1.32 and 0.26, respectively. The amount of nitrogen removal by the anammox process was calculated using the consumption of $NH_4^+$-N. The contribution proportions of the anammox process and the denitrification process of nitrogen removal in the system are presented in Figure 9a, and the flow direction of $NO_2^-$-N is presented in Figure 9b.

The amount and proportion of nitrogen removal by the anammox process in the reactor during Phase 96 kPa, 72 kPa and 65 kPa gradually increased. At 65 kPa, the ratio of $NO_2^-$-N consumed through the anammox process to the theoretical $NO_2^-$-N production was 77.7%, which was 5.6 and 1.5 percentage points higher than that at 96 kPa and 72 kPa, respectively. As air pressure dropped during the nitrogen removal process in the PD/A system, it was seen that anammox had an advantage in the competition with denitrification for $NO_2^-$-N. Consequently, lower air pressure increased the NRE of the PD/A system.

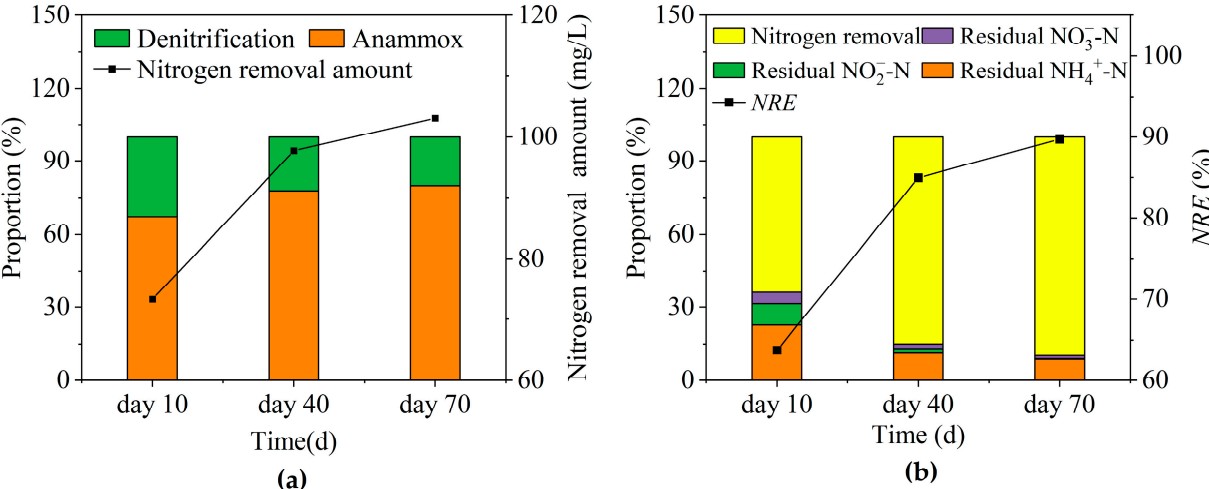

**Figure 9.** Nitrogen removal contribution proportions of the anammox process and the denitrification process (**a**) and the $NO_2^-$-N flow direction (**b**) for the PD/A process in the USR-PD/A system.

### 3.4. Microbial Community Succession in the PD/A System

The composition and relative abundance of the microbial community were analyzed using high-throughput sequencing. Activated sludge samples (day 10, day 40 and day 70) were collected from the PD/A system and were named R.96 kPa, R.72 kPa and R.65 kPa, respectively. Alpha diversity analysis was used to describe the diversity and richness of the activated sludge microbial communities during the three air pressure phases.

The high coverage (>0.98) suggests that deep sequencing can fully express the actual microbial community of all samples. Based on the data of Chao, Shannon and Simpson, the community richness and diversity of sludge in the reactor at the end of the three aerodynamic operation stages were compared (Supplementary Table S3). Compared with those at high air pressure (i.e., 96 kPa), the community richness and diversity of activated sludge under lower pressure (i.e., 72 kPa and 65 kPa) were more significant.

To explore the effect of air pressure on microbial community structure, the distributions of microbial communities were analyzed. This demonstrated that *Proteobacteria* was the dominant phylum at all three pressures, and *Planctomycetes*, *Firmicutes*, and *Bacteroidetes* had relatively high abundances. The distribution of the genus is shown in Figure 10a. *Thauera* has been widely reported to achieve $NO_2^-$-N accumulation, and belongs to the *Proteobacteria* phylum [24,25]. It was the most abundant genus in the USR-PD/A reactor in the R.96 kPa, R.72 kPa and R.65 kPa samples, with proportions of 38.3%, 33.9% and 31.7%, respectively. This was in accordance with the study of Cao [26].

In the PD/A system, there were also a considerable number of heterotrophic denitrification bacteria (e.g., *Pseudoxanthomonas* [27]), which competed with the substrate of partially denitrifying bacteria and decreased its abundance. However, *Thauera* was still the dominant genus. This result was consistent with the stable accumulation of $NH_4^+$-N maintained in the reactor.

*Candidatus Brocadia* and *Candidatus Kuenenia* are common bacteria that can perform anammox functions [28–30]. The relative abundances of anammox bacteria in the R.96 kPa, R.72 kPa and R.65 kPa samples were 5.4%, 7.1% and 7.7%, respectively, which was consistent with the observation that the efficiency of $NH_4^+$-N removal in the reactor increased with lower air pressure. Therefore, air pressure may have an effect on the composition of microbial communities.

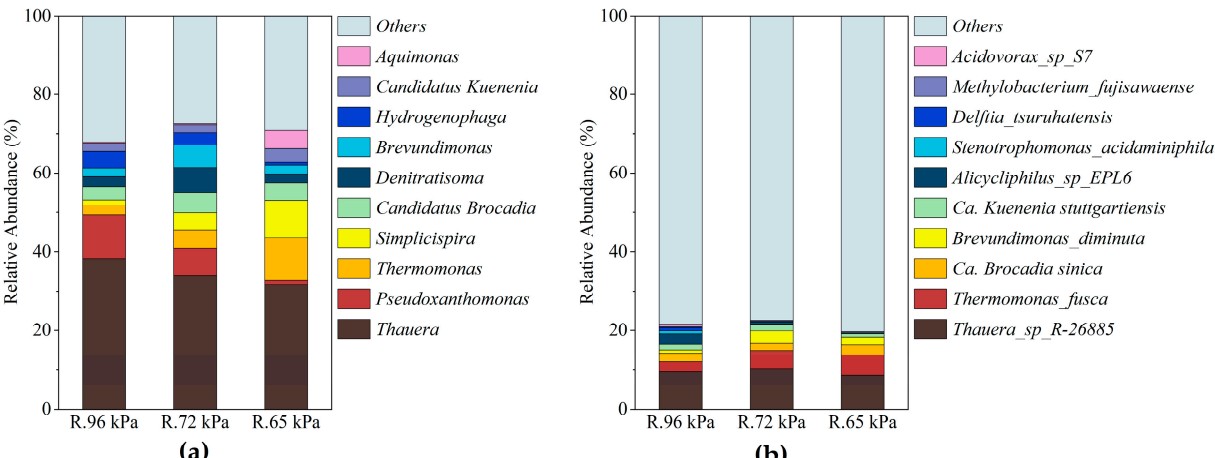

**Figure 10.** The relative abundance of microbial communities at the genus (**a**) and species (**b**) level in the sludge samples collected at different phases. R.96 kPa: sample from Phase 96 kPa (day 10); R.72 kPa: sample from Phase 72 kPa (day 40); R.65 kPa: sample from Phase 65 kPa (day 70).

At the species level (Figure 10b), the relative abundance of *Thauera_sp_R-26885* related to PD was the highest, and the relative abundances in the R.96 kPa, R.72 kPa and R.65 kPa samples were 9.6%, 10.2% and 8.7%, respectively. The anaerobic ammonia-oxidizing bacteria in the samples were Ca. *Brocadiasinica* and Ca. *Kueneniastuttgartiensis* [30], and the relative abundances of these two species in the R.96 kPa, R.72 kPa and R.65 kPa samples were 3.6%, 3.5% and 3.4%, respectively, i.e., almost unchanged. Although the relative abundance was almost the same at different pressures, combined with the higher activity of the anaerobic ammonia-oxidizing bacteria at lower air pressures, this may explain the better $NH_4^+$-N removal efficiency shown by the system at lower air pressure.

## 4. Conclusions

In this study, the influence of low air pressure on the PD process, the anammox process, and the successful combination of anammox and PD into a PD/A process, were investigated. Air pressure itself had little effect on PD. However, the low DO caused by reduced air pressure caused the higher *NAE* in the PD system, reaching values of 78.4 ± 2.8% at 65 kPa. The anammox process was promoted by low air pressure, mainly because low pressure brought higher activity of the anammox bacteria, with *SAA* reaching 26.3 mg·N/(g·VSS·d). The PD/A system achieved higher NRE values at lower air pressures, reaching 93.0 ± 0.3% at 65 kPa. It was found that the contribution proportion of anammox to the nitrogen removal of the system also increased to 77.7%. At the same time, the community structure of the sludge in the PD/A system operated under the three different air pressures was analyzed, and it was found that the *Theura* genus (31.7–38.3%) related to PD was still the largest genus, although its abundance decreased. The relative abundance of genera related to anammox increased (5.4–7.7%), which coincided with the increase in NRE. The PD/A process achieved better NRE under conditions of low air pressure. This study applied the PD/A process in high-altitude regions, providing a sustainable solution for wastewater treatment in high-altitude areas.

**Supplementary Materials:** The following supporting information can be downloaded at: https://www.mdpi.com/article/10.3390/su15139907/s1. Figure S1. Schematic diagram of (A)SBR system (B)USR system; Figure S2. Nitrogen removal performance of SBR-PD during the start-up period; Figure S3. Nitrogen removal performance of PD/A system during the start-up period. Table S1. The composition of synthetic wastewater for (a) SBR-PD (b) SBR-A (c) USR-PD/A; Table S2. The operation conditions of reactors; Table S3. Microbial community richness and diversity of the PD/A system in end stage of different phases. R.96 kPa: sample at Phase 96 kPa (day 10); R.72 kPa: sample at Phase 72 kPa (day 45); R.65 kPa: sample at Phase 65 kPa (day 70).

**Author Contributions:** Conceptualization, Z.H. and G.Z.; data curation, Z.H.; formal analysis, W.D.; investigation, Z.H.; methodology, Z.H.; project administration, G.Z.; resources, Z.H.; supervision, S.L. and G.Z.; validation, Y.L.; visualization, G.Y.; writing—original draft, W.D.; writing—review and editing, Y.L. and G.Z. All authors have read and agreed to the published version of the manuscript.

**Funding:** This research was supported by the National Natural Science Foundation of China (52160004) and Xizang Natural Science Foundation Project of the Xizang Science and Technology Department (XZ202101ZR0092G).

**Institutional Review Board Statement:** Not applicable.

**Informed Consent Statement:** Not applicable.

**Data Availability Statement:** Supplementary Data to this article can be found at (data provided as a Supplementary Materials).

**Conflicts of Interest:** The authors declare no conflict of interest.

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
