# Peer review of "Influence of Low Air Pressure on the Partial Denitrification-Anammox (PD/A) Process"

_sustainability, doi:10.3390/su15139907_

Round 1

Reviewer 1 Report

  1. The main research question is to find out the effect of air pressure fluctuations for nitrogen removal efficiency in the partial denitrification-anammox process; to find out whether the partial denitrification-anammox process is practically implemented in high-altitude regions.
  2. The topic is original and the research is useful in examining the oxygen demand in the wastewater treatment process.
  3. The following observation is important for the subject area: “The partial denitrification-anammox system achieved higher nitrogen removal efficiency values at lower air pressures”.
  4. Research methodology is described well enough. A very big plus would be to test sewage treatment in real conditions using an individual sewage treatment plant.
  5. The conclusions are valid. It was determined that: “The PD/A system achieved higher NRE values at lower air pressures, reaching 93.0 ± 0.3% at 65 kPa’’.
  6. The references are appropriate.
  7. Several inaccuracies have been noted that should be corrected before the article is published:
  • In Figure 5, some series names and designations are not appropriate (eg series 72 kPa NO2—N, (must be NO3-N).
  • Sections of the article should not end with a figure, it is recommended to provide an explanation after the figure.

Author Response

Dear Professor,

Thank you for the thoughtful and thorough feedback on our manuscript. We respond point-by-point to the your comments in the attachment.

We hope the manuscript is now acceptable for publication in Sustainability.

Kindest Regards,

The authors

Reviewer 2 Report

Manuscript Number: sustainability-2434382

Influence of low air pressure on partial denitrifiction-anammox (PD/A) process.

The present work investigates the variation in the performance of a denitrification process combined with an annamox process in order to eliminate the greatest amount of nitrogen from a synthetic wastewater at atmospheric pressure.

GENERAL COMMENTS

The research carried out in this study is very interesting. It is of potential interest for those regions that naturally have reduced atmospheric pressures due to altitude conditions. The combination of both processes has proven to be effective for nitrogen removal.

I think the work is interesting and it should be accepted after some minor revisions.

FORMAT COMMENTS

The title has a mistake, it must be denitrification.

SPECIFIC COMMENTS

I only have some questions for the authors:

-       It is confusing to know when the samples were taken to analyze the structure of the bacterial community in the reactors. In section 2.3 it is indicated that for the USR-PD/A they were taken at the end of the experiment, while section 2.7 indicates that for the PD/A at 10, 40 and 70 days.

-       In addition, as it is written, it seems that for all the experiments samples were taken on those days, however, in Figure 10 it can be seen that for the different pressures, the day of sampling varied. What is this variation due to, why were they not taken on the same day for a better comparison?

-       Some information is missing about the acronyms used in some graphs, for example, in Figure 6 it is not indicated what is kN or kA and in the supplementary material, in figure S2, it is not indicated what is T.

-       On the other hand, during the start-up phase, what was the objective of varying the SRT?

Author Response

(The authors gave the same response as above.)

Reviewer 3 Report

Interesting article, addressing an issue not much covered in the literature and well described. Only Figure 3a is not very clear - maybe it will be better to show each component separately. 

The authors focus on more than just presenting the results but also analyse what causes the phenomena. 

Author Response

(The authors gave the same response as above.)
